# Spectral Analysis of Graph Collapse in Hematopoietic Gene Networks

Seungik Cho
Department of Physics and Astronomy
*Rice University*
Houston, Texas, USA
seungikcho@rice.edu

*Abstract*—Gene regulatory networks (GRNs) orchestrate cell fate decisions, yet conventional transcriptomic analyses often overlook subtle but critical structural disruptions in bioinformatics. We present a spectral framework that reveals local GRN collapse after GATA1 knockout, a key transcription factor in erythroid and eosinophil differentiation. Using Laplacian-based spectral descriptors, we detect a marked collapse in a granulocyte subpopulation, despite minimal global transcriptomic change. This collapse is characterized by low-frequency eigenvalue accumulation, reduced connectivity, and high localized instability. Our findings suggest that GATA1 maintains hidden regulatory attractors in hematopoietic GRNs, and their loss causes lineage-specific structural failure. This is the first application of graph spectral theory to capture cell-type–specific GRN fragility in single-cell perturbation data, offering a theoretical framework for evaluating transcription factor function and cell identity resilience.

*Index Terms*—gene regulatory networks, spectral graph theory, GATA1 knockout, hematopoiesis, Laplacian eigenvalues, single-cell analysis, lineage stability

## I. Introduction

Gene regulatory networks (GRNs) are fundamental to cellular identity and differentiation, enabling cells to maintain lineage fidelity and respond to perturbations through structured transcriptional programs. [1] In hematopoiesis, master transcription factors (TFs) such as GATA1 orchestrate erythroid and eosinophil specification by regulating downstream gene modules. Disruption of these regulators can destabilize these networks, leading to differentiation failure or pathological reprogramming. [2]

With the rise of single-cell RNA-sequencing (scRNA-seq), GRN inference tools have enabled transcriptome-wide reconstruction of regulatory interactions at cellular resolution. According to the study by Gibbs et al. [3], Inferelator 3.0 has been significantly updated to integrate data from distinct cell types, enabling the learning of context-specific regulatory networks and their aggregation into a shared regulatory network. This advancement allows for the integration of large-scale single-cell datasets and the inference of cell-type-specific gene regulatory networks. Similarly, study conducted by Petković et al. [4] shows the usage of GENIE3 to infer GRNs by ranking transcription factor–target relationships based on feature importance scores, enabling scalable inference across thousands of genes. These approaches, however, are often evaluated through gene expression changes, clustering, or regulon scores—focusing on what changes are expressed, not how the network topology itself is disrupted. As a result, subtle but critical structural instabilities within the GRN architecture may be overlooked, particularly in lineage-specific perturbations with minimal transcriptomic footprint.

Spectral graph theory has emerged as a powerful lens to quantify and interpret network topology and could be applied to GRNs. [5] The graph Laplacian matrix, derived from GRN structure, reveals key properties such as connectivity, modularity, and diffusion via its eigenvalue spectrum. [6] Metrics such as algebraic connectivity ($\lambda_1$), spectral entropy, and Laplacian zeta functions capture dynamic robustness and organizational collapse in complex systems [7], yet remain underutilized in GRN-based perturbation studies.

Our study introduces a spectral approach to detect local collapses in GRN structure after transcription factor perturbation, using GATA1 knockout (KO) as a case study. We reconstruct GRNs from hematopoiesis dataset across 24 cellular subpopulations, simulate GATA1 KO using CellOracle pipeline, and extract graph Laplacians for pre- and post-perturbation GRNs. We then analyze eigenvalue spectra using a suite of spectral descriptors. In this work, we propose a novel spectral analysis framework using single-cell GRNs from hematopoietic differentiation data, applying spectral graph theory to quantify TF-induced structural collapse. Our main contributions are summarized as follows:

- We propose a novel computational framework for detecting transcription factor–induced structural collapse in gene regulatory networks using spectral graph theory. Unlike conventional approaches focused solely on gene expression, our method leverages graph-based representations to detect hidden topological vulnerabilities in GRNs.
- We conduct experiments using Laplacian-based spectral metrics—including algebraic connectivity, spectral zeta functions, and graph wavelet energy—to quantify the extent and nature of GRN collapse across distinct cell types. These complementary descriptors allow for a multidimensional characterization of network fragility, capturing both global and localized structural shifts.

## II. METHOD

In this section, we present the methodological foundations of our approach to quantifying GRN collapse under transcription factor perturbation. We begin by reviewing existing statistical and machine learning–based GRN inference methods, including recent spectral approaches. We then formally define the problem of structural collapse in GRNs using Laplacian spectral descriptors. Finally, we introduce our proposed framework, which simulates perturbations, extracts spectral features, and visualizes collapse across clusters using advanced graph-theoretic tools.

### A. Related Works

*1) Classical and Statistical GRN Inference:* Early GRN inference methods relied on statistical co-expression (e.g., correlation matrices, relevance networks) and information-theoretic metrics such as mutual information, as implemented in ARACNE. [8] Regression-based approaches, including LASSO and elastic-net, were used to identify potential regulators per gene. [9] Graphical models, including Bayesian and dynamic Bayesian networks, attempted to model conditional dependencies and temporal dynamics, although their scalability is limited by the computational complexity of inferring directed acyclic graphs (DAGs). [10]

**Fused LASSO for GRN inference.** Let $\mathbf{Y} \in \mathbb{R}^{kN \times 1}$ be the concatenated response vector for a gene across $k$ perturbation experiments, and $\mathbf{X} \in \mathbb{R}^{kN \times kP}$ the corresponding block-diagonal matrix for $P$ transcription factors. Let $\mathbf{W} \in \mathbb{R}^{kP \times kP}$ be a diagonal matrix of biological similarity weights, and $\boldsymbol{\beta} \in \mathbb{R}^{kP \times 1}$ the regression vector. The optimal regulatory program is inferred by solving the following fused LASSO objective:

$$\hat{\boldsymbol{\beta}} = \arg \min_{\boldsymbol{\beta}} \|\mathbf{Y} - \mathbf{X}\boldsymbol{\beta}\|_2^2 + \lambda_1 \|\mathbf{W}\boldsymbol{\beta}\|_1 + \lambda_2 \sum_{i=1}^{k-1} \left\| \boldsymbol{\beta}^{(i)} - \boldsymbol{\beta}^{(i+1)} \right\|_1 \tag{1}$$

where $\lambda_1$, $\lambda_2$ are regularization terms, $\boldsymbol{\beta}^{(i)} \in \mathbb{R}^P$ is the subvector for condition $i$, the first term preserves data fidelity, the second promotes sparsity weighted by biological similarity, and the third enforces inter-condition consistency.

*2) Tree-Based and Ensemble Machine Learning Approaches:* To address scalability, tree-based methods such as GENIE3 decomposed GRN inference into multiple regression tasks using random forest ensembles. By scoring feature importance for each target gene, GENIE3 and its fast variant GRNBoost demonstrated strong performance. [11]

**Regulatory Link Ranking in dynGENIE3.** Let $\mathcal{T}$ be an ensemble of $T$ regression trees trained via random forests to predict the expression of a target gene $j$ from candidate regulators $\{x_i\}_{i \neq j}$, and let $w_{i,j}$ denote the variable importance of regulator $i$ for predicting gene $j$. Then, under the mean decrease impurity criterion, the normalized importance score is defined as

$$\hat{w}_{i,j} = \frac{w_{i,j}}{\sum_{i \neq j} w_{i,j}} \tag{2}$$

where $\hat{w}_{i,j} \in [0,1]$ for all $i \neq j$ and $\sum_{i \neq j} \hat{w}_{i,j} = 1$, the scores $\hat{w}_{i,j}$ are invariant under the addition of uninformative

(noise) features, and if $x_i$ is not used in any tree in $\mathcal{T}$, then $w_{i,j} = \hat{w}_{i,j} = 0$. Hence, the ranking induced by $\hat{w}_{i,j}$ provides a consistent, scale-invariant measure of regulatory influence across targets $j$.

*3) Spectral and Graph-Theoretic GRN Characterization:* Spectral graph theory provides a principled and interpretable framework to analyze the structural integrity of gene regulatory networks. By modeling a GRN as an undirected graph $G = (V, E)$ with adjacency matrix $A$, various spectral features can be computed to characterize topological properties such as cohesion, modularity, and fragmentation. Key metrics include the algebraic connectivity (i.e., second-smallest eigenvalue of the Laplacian), the number of zero eigenvalues (disconnected components), the entropy of the eigenvalue distribution, and the spectral radius. [12]

**Eigenvector Centrality and Node Influence in GRNs.** Let $x \in \mathbb{R}^n$ denote the eigenvector centrality vector of the graph $G$, defined as the solution to the spectral equation:

$$Ax = \rho(A)x \tag{3}$$

where $\rho(A) = \max_{\lambda \in \sigma(A)} |\lambda|$ is the spectral radius of $A$, and $x$ is the corresponding principal eigenvector, normalized such that $\|x\|_2 = 1$. The $i$-th entry $x_i$ represents the centrality of node $v_i \in V$, and satisfies the recursive relation:

$$C_{\text{eiv}}(v_i) \propto \sum_{j \in \mathcal{N}(i)} A_{ij} \, C_{\text{eiv}}(v_j) \tag{4}$$

where $\mathcal{N}(i)$ denotes the set of neighbors of node $v_i$. This formulation highlights that highly central nodes are connected to other central nodes, capturing both direct and indirect regulatory influence.

### B. Problem Definition

Let $\mathcal{G} = \{G^{(c)} = (V, E^{(c)})\}_{c=1}^{C}$ be a set of GRNs corresponding to $C$ distinct cellular subpopulations (clusters) inferred from single-cell RNA sequencing data. Each GRN $G^{(c)}$ shares a common set of genes $V$, but may exhibit different edge sets $E^{(c)}$ based on transcription factor–target relationships. For each $G^{(c)}$, we define the corresponding unnormalized graph Laplacian matrix as:

$$L^{(c)} = D^{(c)} - A^{(c)},$$

where $A^{(c)} \in \mathbb{R}^{N \times N}$ is the adjacency matrix, and $D^{(c)}$ is the degree matrix. We simulate a transcription factor knockout, yielding a perturbed network $\widetilde{G}^{(c)}$ with corresponding Laplacian matrix $\widetilde{L}^{(c)}$. Our objective is to characterize the structural collapse of each GRN under perturbation by analyzing changes in their Laplacian spectra.

### C. Model Architecture

Our proposed framework quantifies structural collapse in GRNs under transcription factor perturbations (e.g., GATA1 knockout) using spectral graph theory. As illustrated in Figure 1, the framework constructs GRNs from scRNA-seq data via the CellOracle pipeline, generating both wild-type and perturbed networks. These are then transformed into graph

Laplacians, from which a spectral descriptor module computes key features of the eigenvalue spectrum—such as algebraic connectivity, number of zero eigenvalues, and spectral entropy.

To characterize GRN collapse in a multidimensional space, we perform differential spectrum analysis and incorporate additional tools including the spectral zeta function, spectral graph wavelet energy, and Wasserstein distances between spectral distributions. The entire pipeline is implemented in Python and the steps are following:

*a) Spectral Descriptor Data Generation:* For each GRN $G^{(c)}$ before and after perturbation, we compute the unnormalized graph Laplacian $L = D - A$ and extract its number of genes acting as nodes in the graph, number of edges, and eigenvalue spectrum $\{\lambda_i\}_{i=1}^{p}$. Each data of before GATA1 KO and after GATA1 KO were stored as csv file. We focus on four key spectral metrics: (i) Algebraic connectivity ($\lambda_1$), (ii) Number of zero eigenvalues, (iii) Mean and variance of the spectrum, (iv) Spectral entropy (disorder of eigenvalue distribution).

*b) Differential Spectrum Analysis:* For each cluster, we quantify the impact of perturbation by computing the difference between perturbed and baseline values for key spectral metrics. For example, we calculate the change in algebraic connectivity as the perturbed value minus the baseline value. From there, we extract the change on value of algebraic connectivity and the number of zero eigenvalues to evaluate the collapse.

*c) 3D Collapse Manifold Visualization:* To illustrate the multi-dimensional nature of collapse, we embed clusters in a 3D space defined by ($\Delta\lambda_1, \Delta$Variance, $\Delta$Entropy). This manifold reveals localized perturbation-sensitive states.

*d) Spectral Zeta Function:* To detect low-frequency compression, we compute the spectral zeta function $\zeta(s) = \sum_{i=1}^{N'} \lambda_i^{-s}$ over the nonzero Laplacian eigenvalues $\lambda_i$, with $s \in \{1, 2, 3\}$ controlling sensitivity to small eigenvalues. Larger $s$ amplifies the influence of low-frequency modes, making $\zeta(s)$ a proxy for spectral skewness.

*e) Spectral Graph Wavelet Energy:* We computed wavelet energy $\sum_i g^2(s\lambda_i)$ using a Gaussian kernel $g(\lambda) = \exp(-(s\lambda)^2)$ to assess signal propagation capacity across scales. We used $s \in \{0.1, 1, 100\}$ to capture local (node-level), meso-scale (modular), and global (network-wide) diffusion dynamics, enabling localization of structural collapse in frequency space.

*f) Spectral Wasserstein Distance:* Finally, we compute the Wasserstein distance between spectral distributions before and after perturbation, using exported eigenvalue statistics, to quantify global structural reorganization across clusters.

## III. EXPERIMENTS

In this section, we empirically validate our spectral graph framework for quantifying GRN collapse under transcription

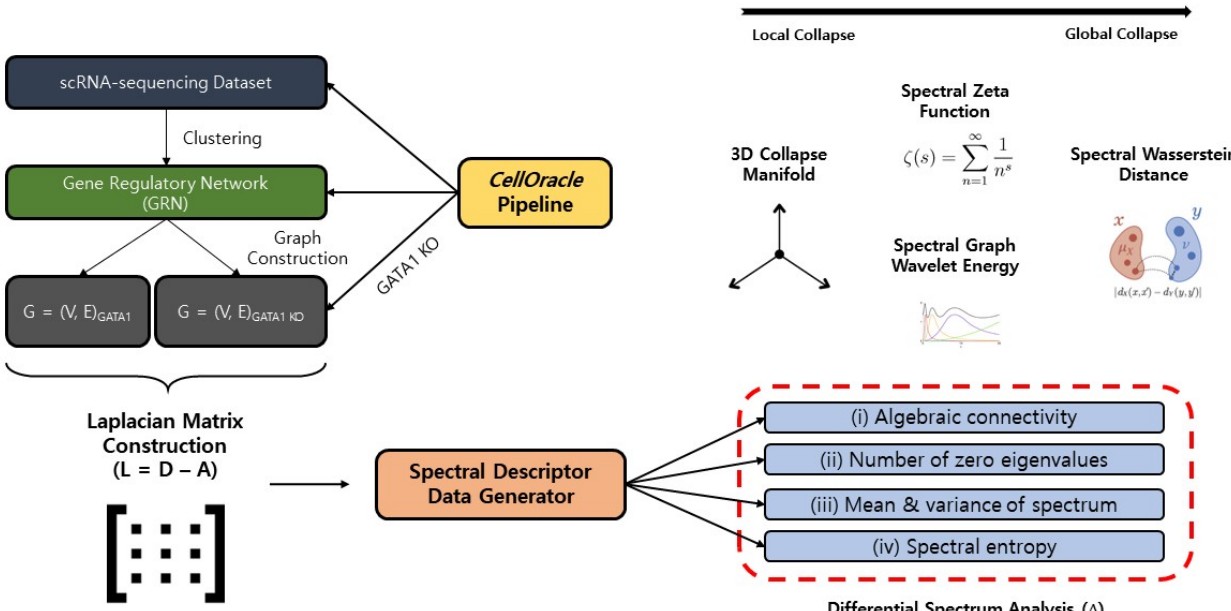

Fig. 1. Overview of our spectral graph framework for quantifying structural collapse in GRNs under transcription factor perturbation. Starting from a scRNA-seq dataset, we construct wild-type and GATA1 knockout GRNs using the CellOracle pipeline, convert them into Laplacian matrices, and compute key spectral descriptors including algebraic connectivity, number of zero eigenvalues, spectral entropy, and spectrum statistics. These features define a collapse manifold. We additionally apply spectral zeta functions, wavelet energy, and Wasserstein distance to quantify the extent of local-to-global GRN reorganization.

factor perturbations. We begin by describing the dataset and GRN construction process, including in silico GATA1 knockout simulations. We then detail the implementation of our spectral evaluation pipeline and present visualization strategies for interpreting the structural reorganization of networks across cell populations.

### A. Dataset and GRN Construction

We utilize the single-cell RNA-sequencing dataset published by Paul et al., which provides a high-resolution transcriptional map of early hematopoietic differentiation in mouse bone marrow. This dataset contains gene expression profiles of 2,671 lineage-negative progenitor cells across 1,999 genes, measured using MARS-seq. In addition to raw expression, the dataset includes annotations such as pseudotime trajectories, dimensionality-reduced coordinates, and cluster-based lineage identities. [13] We accessed this preprocessed dataset via the CellOracle framework using the co.data.load_Paul2015_data() utility, which loads normalized expression matrices and metadata. Cells were grouped into 24 transcriptionally distinct subpopulations based on the louvain_annot field, capturing key hematopoietic trajectories including erythroid (Ery), granulocytic (GMP/GMPI/Gran), monocytic (Mo), and megakaryocytic (MEP/Mk) branches. [14]

For each subpopulation (cluster), we constructed a context-specific GRN by integrating scRNA-seq expression profiles with a genome-wide transcription factor binding prior from scATAC-seq data. This prior was loaded via load_mouse_scATAC_atlas_base_GRN(). GRN inference was performed using ridge-regularized regression, treating TFs and their targets as nodes in a directed, weighted bipartite graph. To simulate transcriptional perturbation, we conducted an in silico knockout of GATA1—a key regulator of erythroid fate—by setting its expression to zero prior to GRN inference. This generated perturbed GRNs for each cluster. Both baseline and perturbed networks were exported for spectral analysis, enabling downstream quantification of collapse severity across cell states.

### B. Spectral Collapse Evaluation Pipeline

We summarize the proposed GRN collapse quantification framework in Algorithm 1. Given the GRNs before and after perturbation, we compute Laplacian spectra, extract key descriptors, and quantify structural shifts using multiple spectral features.

### C. Spectral Visualization

To intuitively interpret the structural changes in GRNs, we visualize the extracted spectral descriptors using several 1D and 3D methods. Barplots are used to highlight per-cluster differences in algebraic connectivity ($\Delta\lambda_1$), zero eigenvalue count, and spectral entropy. To explore higher-dimensional perturbation patterns, we embed each cluster into a 3D manifold defined by $(\Delta\lambda_1, \Delta\text{Variance}, \Delta\text{Entropy})$ using matplotlib. Additionally, exponential decay curves $f(t) = \exp(-\lambda_1 t)$ are plotted for each cluster to simulate dynamical stability. We

---

**Algorithm 1** Spectral Collapse Quantification for GRNs
___
**Require:** Baseline and perturbed GRNs $\{G^{(c)}, \widetilde{G}^{(c)}\}_{c=1}^{C}$
1: **for** each cluster $c = 1$ to $C$ **do**
2:     Construct symmetric adjacency matrices:
        $A^{(c)} = \frac{1}{2}(A^{(c)} + (A^{(c)})^{\top}), \widetilde{A}^{(c)} = \frac{1}{2}(\widetilde{A}^{(c)} + (\widetilde{A}^{(c)})^{\top})$
3:     Compute Laplacians:
        $L^{(c)} = D^{(c)} - A^{(c)}, \widetilde{L}^{(c)} = \widetilde{D}^{(c)} - \widetilde{A}^{(c)}$
4:     Compute eigenvalue spectra:
        $\{\lambda_i^{(c)}\} = \text{eig}(L^{(c)}), \{\widetilde{\lambda}_i^{(c)}\} = \text{eig}(\widetilde{L}^{(c)})$
5:     Extract descriptors: algebraic connectivity $\lambda_1$, zero eigenvalue count $Z$, spectral entropy $H$, mean and variance
6:     Differential analysis:
        $\Delta\lambda_1^{(c)} = \widetilde{\lambda}_1^{(c)} - \lambda_1^{(c)}$
        $\Delta Z^{(c)} = \widetilde{Z}^{(c)} - Z^{(c)}$
        $\Delta H^{(c)} = \widetilde{H}^{(c)} - H^{(c)}$
7:     Metrics:
        Zeta: $\zeta^{(c)}(s) = \sum_i \lambda_i^{-s}$ for $s = 1, 2, 3$
        Wavelet Energy: $E^{(c)}(s) = \sum_i \exp(-s^2\lambda_i^2)$
        Wasserstein: $W^{(c)} = \mathcal{W}_1(\{\lambda_i^{(c)}\}, \{\widetilde{\lambda}_i^{(c)}\})$
8: **end for**
9: **return** $\{\Delta\lambda_1^{(c)}, \Delta Z^{(c)}, \Delta H^{(c)}, \zeta^{(c)}, E^{(c)}, W^{(c)}\}_{c=1}^{C}$
___

further visualize spectral wavelet energy across multiple scales and compute Wasserstein distance matrices to quantify global spectral shifts across clusters.

## IV. RESULT ANALYSIS

We now present a detailed analysis of GRN structural responses to GATA1 knockout, using our proposed spectral graph framework. By leveraging multiple spectral descriptors, we characterize both global and local network perturbations across hematopoietic cell states.

### A. Differential Spectrum Analysis

To uncover cluster-specific structural disruptions after GATA1 knockout, we begin by analyzing two complementary spectral metrics: the number of zero eigenvalues and the algebraic connectivity ($\lambda_1$) of the cluster-wise graph Laplacians. These indicators respectively quantify local fragmentation and global connectivity of GRNs. In Figure 2, we visualize the change in the number of zero eigenvalues ($\Delta\#(\lambda = 0)$) for each cluster after GATA1 perturbation. As expected, many erythroid (Ery) clusters exhibit substantial increases in disconnected components, reflecting their biological dependence on GATA1. For instance, Ery_5, Ery_6, Ery_8, Ery_9 show elevated $\Delta\#(\lambda = 0) > 60$, indicating strong fragmentation.

However, Gran_3 displays the largest increase in zero eigenvalues ($+79$), surpassing all erythroid clusters, despite not being a canonical GATA1 target. This suggests that Gran_3 undergoes significant internal disintegration even though it is not globally disconnected in the same way. To further investigate this anomaly, we turn to Figure 3, which shows the change in algebraic connectivity ($\Delta\lambda_1$) for each cluster. This metric quantifies the strength of global connectivity and

spectral cohesion. Here, Gran_3 exhibits only a mild decrease in $\lambda_1$ (approximately $-0.0059$), indicating that the overall graph structure remains weakly connected from a global spectral perspective.

This apparent contradiction—low $\Delta\lambda_1$ but high $\Delta\#(\lambda = 0)$—points to a phenomenon we refer to as hidden fragmentation or local collapse. In Gran_3, the GRN retains superficial connectivity while internally fragmenting into disconnected components, which is invisible to global descriptors alone.

These findings motivate deeper inspection of Gran_3 using higher-order descriptors (e.g., spectral entropy, wavelet energy, Wasserstein distance), which we explore in the subsequent sections.

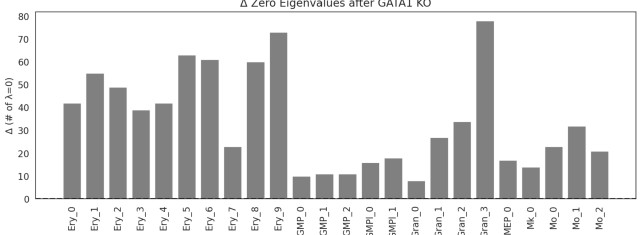

Fig. 2. Change in number of zero eigenvalues ($\Delta\#(\lambda = 0)$) after GATA1 KO. Notably, Gran_3 shows the largest increase, indicating severe local fragmentation.

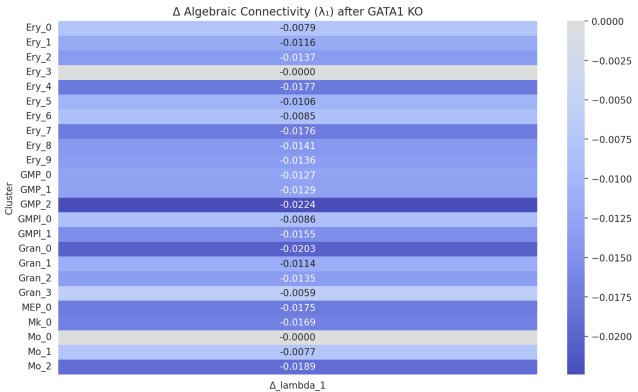

Fig. 3. Change in algebraic connectivity ($\Delta\lambda_1$) after GATA1 KO. Gran_3 shows only mild decrease, revealing hidden fragmentation undetectable by global metrics.

### B. Spectral Collapse Manifold Analysis

In Figure 4, we project each cell cluster into a three-dimensional spectral space spanned by the change in algebraic connectivity ($\Delta\lambda_1$), Laplacian variance ($\Delta$Variance), and spectral entropy ($\Delta$Entropy) after GATA1 knockout. This manifold enables intuitive visual detection of clusters that exhibit structurally significant shifts in their gene regulatory networks. Among all clusters, Ery_5, Ery_7, and Ery_3 occupy the upper region of the entropy axis, indicating a noticeable increase in spectral entropy following perturbation. This suggests that these erythroid subclusters exhibit elevated topological disorder or diffuseness, consistent with disrupted

regulatory coherence upon GATA1 deletion. Their $\Delta\lambda_1$ values are also moderately negative, reinforcing the notion of reduced global connectivity.

Conversely, Gran_3 is characterized by a relatively low $\Delta$Entropy and moderately small $\Delta\lambda_1$, indicating a subtle form of localized collapse. Although its global entropy remains low, its position near the lower entropy plane suggests that Gran_3 retains ordered structure in aggregate but may conceal discrete vulnerabilities, aligning with the increase in zero eigenvalues previously observed in Figure 2. Additionally, most myeloid clusters (Mo_0, Mo_1, Mk_0) cluster near the origin, suggesting minimal perturbation impact across all three spectral dimensions, consistent with their non-erythroid lineage identity.

Taken together, this visualization highlights distinct structural responses to GATA1 knockout. Erythroid clusters show marked entropy increases indicative of regulatory breakdown, while Gran_3 reveals a unique regime of low entropy but latent collapse, warranting further localized analysis.

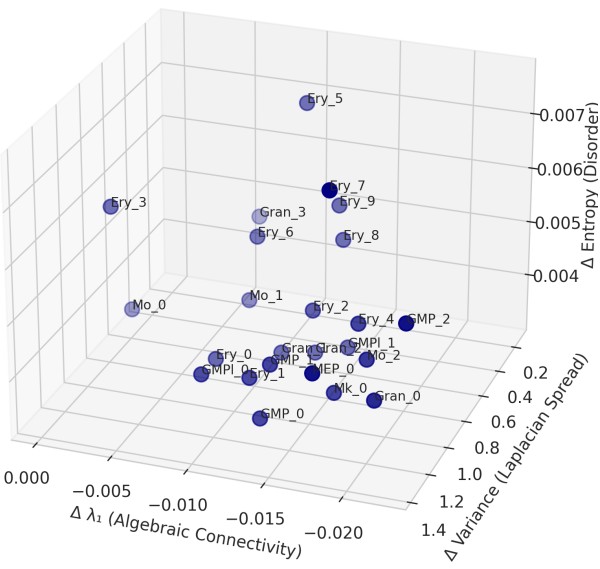

Fig. 4. Three-dimensional manifold visualization of cluster-specific spectral collapse after GATA1 knockout. Each point represents a cell cluster, embedded in a space defined by the change in algebraic connectivity ($\Delta\lambda_1$), Laplacian variance ($\Delta$ Variance), and spectral entropy ($\Delta$ Entropy). Erythroid clusters such as Ery_5, Ery_7, and Ery_3 exhibit significant increases in entropy, indicating disordered GRN topology. In contrast, Gran_3 lies in a region of low entropy and moderate collapse, suggesting a distinct mode of hidden structural vulnerability.

### C. Spectral Zeta Function Analysis

To quantify cluster-wide spectral reconfiguration following GATA1 knockout, we compute the spectral zeta function $\zeta_s = \sum_i \lambda_i^{-s}$ with exponent $s = 2$. This function aggregates contributions from all non-zero eigenvalues of the Laplacian matrix, with higher sensitivity to lower eigenvalues—thereby

capturing large-scale structural distortions not visible in traditional global metrics like algebraic connectivity.

As illustrated in Figure 5, the change $\Delta\zeta_2 = \zeta_2^{\text{post}} - \zeta_2^{\text{pre}}$ varies across clusters. Most strikingly, Gran_3 exhibits the largest increase in $\zeta_2$, far surpassing all other clusters. This dramatic shift suggests a localized yet distributed spectral collapse, where small structural perturbations across the network collectively induce a global reorganization of the Laplacian spectrum.

This observation is consistent with our earlier findings: while Gran_3 did not show prominent change in algebraic connectivity (Figure 3), it harbored a sharp increase in zero eigenvalues—indicating silent fragmentation at the local scale. The surge in $\zeta_2$ now confirms that these local disconnections compound into a latent yet significant form of collapse, one that is distributed throughout the spectrum rather than concentrated at a single structural scale.

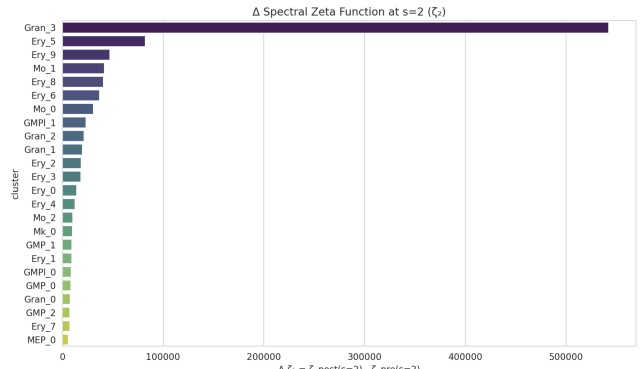

Fig. 5. Change in spectral zeta function $\zeta_2$ after GATA1 knockout. Gran_3 shows the most prominent increase in $\Delta\zeta_2$, indicating a hidden collapse not captured by low-order metrics. This suggests a distributed spectral reconfiguration across the entire Laplacian eigenvalue spectrum.

### D. Spectral Wavelet Energy Analysis

To dissect the scale-dependent effects of GATA1 perturbation, we applied Spectral Graph Wavelet Transform (SGWT) to compute wavelet energy distributions across different diffusion scales. This method captures how effectively each cluster's GRN transmits regulatory signals at various scales.

As illustrated in Figure 6, most clusters exhibit a noticeable decline in wavelet energy as the scale increases, indicating weakened signal propagation over longer distances. However, Gran_3 stands out by maintaining relatively high energy even at the largest scale $s = 100$, suggesting enhanced capacity to preserve mid- to long-range signal integrity.

This observation reinforces our earlier finding that while Gran_3 shows spectral irregularities in global descriptors such as the zeta function, it simultaneously retains strong structural coherence across scales. The cluster's resistance to wavelet energy dissipation implies a form of spectral resilience that differentiates it from truly collapsed networks like Ery_5 or Mo_1, which exhibit steep spectral decay.

Therefore, rather than undergoing a complete breakdown, Gran_3 appears to reorganize in a way that preserves signal

propagation pathways, highlighting a localized and functionally adaptive response to perturbation.

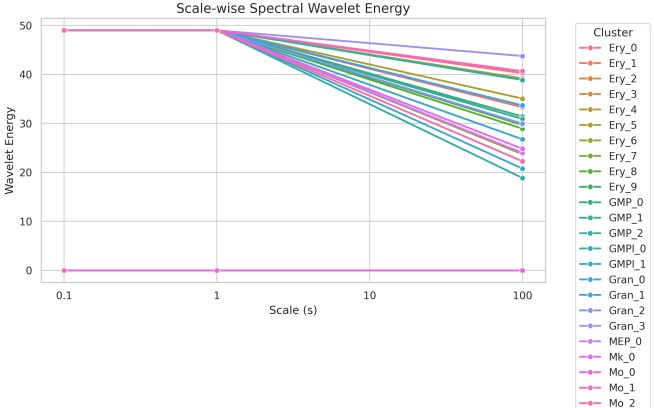

Fig. 6. Scale-wise spectral graph wavelet energy for each GRN cluster. While most clusters show monotonic decay of energy with increasing scale, Gran_3 maintains high wavelet energy at large scales ($s = 100$), suggesting preserved signal propagation and structural resilience following perturbation.

### E. Global Spectral Collapse Analysis

To quantify the global structural reorganization of GRNs induced by GATA1 knockout, we compute the Wasserstein distance between spectral feature vectors before and after perturbation for each cluster. This analysis complements local metrics, such as algebraic connectivity and zero eigenvalues, by capturing the aggregated shift in spectral geometry. A higher Wasserstein distance indicates a more pronounced global alteration in the GRN structure following perturbation.

Notably, clusters such as Ery_7, MEP_0, and Ery_4 exhibit the largest spectral divergence, consistent with erythroid lineage being highly responsive to GATA1 disruption. In contrast, Gran_3, which was previously identified as collapse-prone based on local metrics (i.e., low $\Delta\lambda_1$ and an increase in the number of zero eigenvalues), ranks among the lowest in Wasserstein distance. This shows that local collapses may occur without globally noticeable perturbations in spectral structure.

Together, this result underscores the importance of combining both local and global spectral descriptors to uncover lineage-specific vulnerability. While global metrics like Wasserstein distance reveal large-scale network remodeling, only fine-grained local analysis exposes subtle, localized instability as seen in Gran_3.

### V. CONCLUSION

This study presents a novel theoretical framework for quantifying TF perturbations on GRNs using spectral graph theory. Unlike conventional expression-based analyses, our method captures latent topological shifts by modeling GRNs as Laplacian graphs and decomposing their eigenvalue spectra.

While GNNs and diffusion-based models perform well predictively, they often act as black boxes, limiting mechanistic interpretability. Our method instead uses analytically

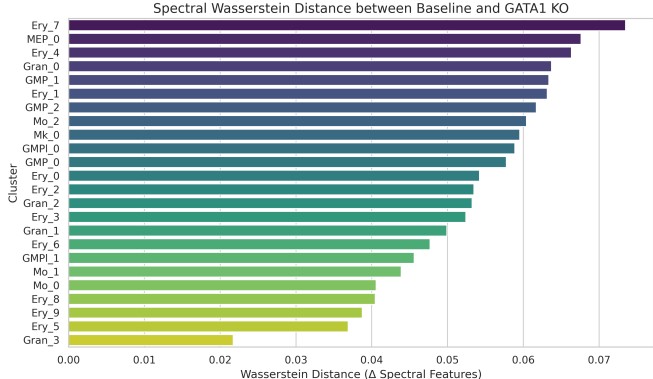

Fig. 7. Spectral Wasserstein distance between baseline and GATA1 KO GRNs across clusters. Higher values indicate larger global divergence in the spectral structure.

grounded, interpretable spectral descriptors to directly quantify topological collapse. These metrics are explainable and trace back to specific structural changes such as fragmentation or signal dissipation.

A central finding is the detection of significant collapse within the granulocytic cluster upon GATA1 knockout. Despite appearing stable in global comparisons such as low Wasserstein distance, this cell state exhibited one of the largest local disruptions in both algebraic connectivity and spectral entropy—indicating a loss of modular cohesion and signaling bottlenecks. This coexistence of global stability and local fragmentation emerges as a key signature of the GATA1 perturbation effect. These theoretical results corroborate recent biological observations that GATA1 depletion selectively perturbs granulocytic differentiation pathways. For instance, according to Harper et al. [15], GATA1 was related with both the yield and maturation of neutrophils derived from human pluripotent stem cells.

While this study focuses on GATA1 as an erythroid regulator, our framework is readily applicable to other transcription factors. Future work will extend this approach to perturbations such as SPI1 and CEBPA, which govern myeloid and granulocytic differentiation, respectively. In addition, incorporating edge directionality—through normalized random walk Laplacians or other asymmetric operators—may capture causal regulatory flows and further enhance the biological realism of inferred collapse patterns. Together, these extensions will enable systematic quantification of lineage-specific GRN vulnerabilities across diverse transcriptional landscapes. Moreover, we envision its clinical utility in early diagnostics and perturbation simulation. For instance, in cancer or hematological disease models where TF perturbation is known, applying spectral GRN descriptors could help identify latent collapse or lineage drift at early stages.

## Acknowledgment

I would like to thank the developers of CellOracle—particularly Dr. Samantha A. Morris and her team at the Department of Developmental Biology and Genetics, Washington University School of Medicine in St. Louis—for providing a powerful and accessible framework for gene regulatory network inference and perturbation simulation. Their open-source tool was instrumental in enabling the analyses presented in this study.

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
