# OpenReview forum: "Spectral Analysis of Graph Collapse in Hematopoietic Gene Networks"
_IEEE.org/EMBS/BHI/2025/Conference — BHI 2025_

### Official Review · Reviewer_9qux · 2025-07-15
**Review: Spectral Analysis of Graph Collapse in Hematopoietic Gene Networks**

**Confidence:** 3
**Clarity Of Writing:** good
**Clinical Significance:** good
**Methodological Novelty:** good
**Overall Rating:** 7

**Experiments And Results:**

good

**Questions For The Authors:**

1. Provide more context or hypotheses for why Gran 3 may be affected by GATA1 KO, even if it's not a canonical target.

2. Compare performance or interpretability with graph neural networks or diffusion-based models for perturbation analysis.

3. Discuss how incorporating edge directionality (e.g., via directed Laplacians or random walk Laplacians) might enhance biological realism.

4. Include statistical tests or confidence intervals for major spectral changes to assess their robustness.

5. Test the method on other perturbations (e.g., SPI1 or CEBPA KO) to evaluate generality of the framework.

**Strengths:**

1. The integration of spectral graph theory into the analysis of GRNs under perturbation is novel and provides a principled way to quantify network fragility.

2. The identification of a “hidden collapse” in the granulocyte cluster (Gran 3) after GATA1 knockout adds biological plausibility and potentially new insight into transcriptional regulation mechanisms.

3. The use of complementary descriptors—ranging from local (algebraic connectivity, zero eigenvalues) to global (spectral entropy, zeta functions, Wasserstein distance)—provides a rich, multi-scale characterization of network reorganization.

4. The 3D collapse manifold and spectral energy plots are effective at conveying the complexity of GRN disruption, especially in distinguishing between local and global collapse.

5. The authors provide detailed methodological steps and implementation notes, including use of public datasets and open-source tools (e.g., CellOracle), enhancing reproducibility.

**Summary Of The Paper:**

This manuscript introduces a novel spectral graph-theoretic framework for identifying structural collapses in gene regulatory networks (GRNs) following transcription factor perturbation, with a specific focus on GATA1 knockout in hematopoietic differentiation. By leveraging Laplacian spectral descriptors (e.g., algebraic connectivity, spectral entropy, wavelet energy, and zeta functions), the authors demonstrate how subtle, localized disruptions in GRN topology can be revealed—even in the absence of major transcriptomic shifts. The study finds that granulocyte subpopulation Gran 3 undergoes a latent, non-obvious collapse after GATA1 knockout, which evades detection by conventional global metrics but becomes apparent through spectral decomposition.

**Weaknesses:**

1. While Gran 3 is highlighted as a site of hidden structural collapse, the biological implications of this finding (e.g., how it affects lineage specification or downstream gene expression) are speculative and not experimentally validated.

2. The authors mention that classical machine learning methods lack interpretability, but no direct comparisons to more recent interpretable deep learning or graph neural network approaches are presented.

3. The framework involves several hyperparameters (e.g., scale of wavelet transforms, s in zeta functions) without a thorough sensitivity or ablation analysis, which limits the robustness assessment.

---

### Official Review · Reviewer_dmyJ · 2025-07-17
**Innovative theoretical approach of graph spectral theory with future potential in clinical practice**

**Confidence:** 5
**Clarity Of Writing:** good
**Clinical Significance:** fair
**Methodological Novelty:** good
**Overall Rating:** 7

**Experiments And Results:**

fair

**Questions For The Authors:**

* presentation of SOTA could be extended, updating references and introducing limitations and challenges with respect to the examined scientific field
* how could proposed approach be applied in clinical scenarios? How far is the transformation of "theoretical proof-oc-concept" to a ready-to-use framework. What kind and amount of data are required towards this direction?
* comparative results against existing approaches could be considered
* quantitative assessments using statistical significance metrics could be included
* how do you address the case that uncertainties in GRN inference might affect spectral analysis results?
* please provide more details on the approximation method for the zeta function calculation
* please describe in short the motivation beyond the specific selection of scales (0.1, 1, 100) for wavelet analysis

**Strengths:**

* clear demonstration of paper novelty and contribution
* description of methods is supported by proper mathematical background
* detailed presentation of implementation framework
* promising preliminary results

**Summary Of The Paper:**

This study presents a spectral analysis framework to detect structural collapse in gene regulatory networks following transcription factor perturbations, which constitutes the first application of graph spectral theory to capture cell-type–specific GRN fragility in single-cell perturbation data.

**Weaknesses:**

* presentation of SOTA is short and not based on the latest advances/references
* theoretical approach, applicability in clinical practice is not clear to the reader
* validation is performed on a limited dataset
* no comparative results against reference approaches

---

### Official Review · Reviewer_1XDv · 2025-07-18
**Spectral Analysis of Graph Collapse in Hematopoietic Gene Networks**

**Confidence:** 1
**Clarity Of Writing:** good
**Clinical Significance:** good
**Methodological Novelty:** fair
**Overall Rating:** 4

**Experiments And Results:**

poor

**Questions For The Authors:**

1. Can you provide experimental evidence that Gran 3 cells with predicted spectral collapse show measurable functional deficits?

2. How do you determine statistical significance of spectral changes?

3. For parameters and sensitivity perspective, spectral metrics may be artifacts of GRN thresholds. Results could be method-dependent rather than biologically meaningful.

4. How do you maintain global connectivity with increased local fragmentation?

5. How sensitive are your spectral metrics to GRN construction parameters?

**Strengths:**

The authors employ multiple complementary spectral descriptors.

The identification of fragmentation in Gran 3 is a particularly valuable insight, where local disconnection occurs without global connectivity loss.

The focus on GATA1 in hematopoiesis is well-chosen, and methodology is clearly described with algorithmic details.

**Summary Of The Paper:**

This paper introduces a spectral graph theory framework to detect structural collapse in gene regulatory networks (GRNs).
The key finding is the detection of significant local collapse in granulocyte cluster Gran 3

**Weaknesses:**

1. The paper lacks experimental validation of the predicted structural collapse. There is no demonstration that the spectral changes correspond to measurable biological phenotypes

2. The paper provides insufficient analysis of how sensitive the spectral metrics are.

3. The analysis is restricted to one transcription factor (GATA1) in one biological system (hematopoiesis).   How the general approach across different TFs and tissues can be addressed.

4. There is no statistical framework for determining when spectral changes are significant versus noise.